# The Burning Pain Transcriptome in the Mouse Primary Somatosensory Cortex

**DOI:** 10.3390/ijms26083538

**Published:** 2025-04-09

**Authors:** Virág Erdei, Zoltán Mészár, Angelika Varga

**Affiliations:** 1Department of Anatomy, Histology and Embryology, Faculty of Medicine, University of Debrecen, H-4032 Debrecen, Hungary; erdei.virag6@gmail.com (V.E.); meszarz@anat.med.unideb.hu (Z.M.); 2Department of Radiology, Central Hospital of Northern Pest—Military Hospital, Budapest, H-1134 Budapest, Hungary

**Keywords:** burn injury, primary somatosensory cortex, RNA sequencing, inflammatory pain, retrograde endocannabinoid signaling

## Abstract

Our previous research has demonstrated that the spinal cord undergoes epigenetic and molecular alterations following non-severe burn injury (BI). However, the primary somatosensory cortex (S1), crucial for pain perception, remains unexplored in this context. Here, we investigated transcriptomic alterations in the S1 cortex of mice subjected to BI or formalin application (FA) to the hind paw, utilizing RNA sequencing (RNA-seq) one hour after injury. RNA-seq identified 1116 differentially expressed genes (DEGs) in BI and 136 DEGs in formalin-induced inflammatory pain. Notably, 82.4% of DEGs in BI and 32.4% in FA were downregulated. A total of 42 upregulated and 17 downregulated overlapping DEGs were identified, indicating significant differences in the cortical processing of pain based on its origins. Gene Ontology analysis reveals that BI upregulated mitochondrial functions and ribosome synthesis, whereas axon guidance, synaptic plasticity, and neurotransmission-related processes were downregulated. By contrast, formalin treatment mainly impacted metabolic processes. Kyoto Encyclopedia of Genes and Genomes (KEGG) pathway analysis highlights the significance of retrograde endocannabinoid signaling (REC) in the response to burn injury. These findings demonstrate that transcriptomic remodeling in the S1 cortex is dependent on the sensory modality and suggest that the REC network is activated during acute pain responses following BI.

## 1. Introduction

A burn injury (BI) can significantly reduce the quality of life, leading to lasting changes not only at the site of the injury site but also resulting in pain and neurological alterations due to chronic neuroplasticity [1,2,3,4]. Various brain regions, including the anterior cingulate cortex (ACC), insular cortex, medial prefrontal cortex, and primary and secondary somatosensory cortices (S1 and S2), are involved in the experience of pain [5,6,7], even though they are temporally and spatially distant from the initial injury. Epidemiological data show that burn patients, whether they have severe burns (≥20% total body surface area (TBSA)) or non-severe burns (<20% TBSA), require extended hospital stays due to neurological dysfunction [2,4]. Despite the functional and structural changes observed in the central nervous system (CNS) of burn patients [1,2,3,4], our understanding of the molecular basis underlying these burn-induced functional consequences remains limited.

The sensory cortex, particularly the primary somatosensory cortex, plays a crucial role in encoding the sensory-discriminative aspects of noxious input, such as identifying the location of the pain stimulus and the perceptual discrimination of pain intensity [8,9,10]. Currently, a PubMed search for RNA sequencing (RNA-seq), somatosensory cortex, and pain reveals only four publications [11,12,13,14]. These studies investigate various models of neuropathic pain, including tibial nerve injury and infraorbital nerve transection [12,13]. In addition, other studies have demonstrated a significant association between specific chronic pain models (e.g., spared nerve injury, chronic constriction injury, postherpetic neuralgia, and tibial nerve damage) and transcriptional changes in various brain regions, including the ACC, prefrontal cortex, and S1 cortex [5,6,15,16,17,18]. Although the neurobiological aspects of nociception are well-established [19], the molecular mechanisms underlying pain processing in the somatosensory cortex, particularly in relation to burn-induced pain, remain poorly understood.

Our recent findings reveal significant differences in the central processing of pain models of various origins at the spinal cord level [20]. This suggests that the transcriptomic patterns in the S1 cortex may vary between the burn injury model and the inflammatory pain model induced by formalin treatment (FA). To test this hypothesis, we utilized next-generation RNA-seq to examine alterations in gene expression in the S1 cortex of adult male mice one hour after a non-severe burn injury or formalin application to the hind paw. We compared the gene expression patterns from the burn injury to those induced by formalin-induced inflammatory pain models in adult male mice. It is important to note that, due to ethical considerations concerning animal welfare, the survival time was limited to one hour post-injury.

This comparative analysis allowed us to identify differentially expressed genes (DEGs) and delineate key signaling pathways and functional gene clusters through Gene Ontology (GO) enrichment analysis and Kyoto Encyclopedia of Genes and Genomes (KEGG) Pathway analysis during the acute phase of noxious stimuli. Additionally, immunofluorescent studies targeting the CB1 receptor and Ggamma14 were conducted to validate the gene expression results.

## 2. Results

### 2.1. Burn Injury Significantly Alters the Activity of Different Cortical Neuron Populations in the S1 Cortex, as Assessed by c-Fos Immunohistochemistry

Neuronal activation was investigated in the S1 cortex following burn injury using c-Fos immunohistochemistry. c-Fos is an immediate early gene and a well-known marker for neuronal activation [21,22]. Its expression along the pain pathway is often utilized to study immediate pain responses [21,22]. Therefore, comparing c-Fos expression between the control and burn injury-treated samples was a logical first step (Figure 1a).

c-Fos expression in the spinal cord typically increases rapidly after a painful stimulus, peaking at 1 to 2 h and returning to baseline within 4 to 6 h. This expression is elevated in specific regions of the CNS, including the S1 cortex, in response to neuropathic or inflammatory pain [21,22,23]. In murine models, the medial aspect of the S1 cortex is associated with the representation of the hind paw region [24]. A schematic representation of the coronal section of the mouse brain, illustrating the S1 cortical area, is depicted in Figure 1b. Figure 1c displays representative confocal images of c-Fos immunoreactivity (IR) from a control mouse and those with burn injury. After burn injury, c-FOS-IR exhibited a notable increase, primarily concentrated in layers 2/3 on both sides of the S1 cortex when compared to control samples. Furthermore, sporadic c-Fos-IR was observed in the deeper layers (layers 5/6). In the region of the S1 cortex that represents the lower limb, 28 out of 4858 DAPI-labeled cell nuclei exhibited c-Fos-IR, accounting for 0.58 ± 0.05% (from 10 slices of 4 mice). In the contralateral S1 cortex, 1.8 ± 0.26% of DAPI-labeled cells displayed c-Fos-IR after a burn injury, with an average count of 97 out of 5294 cells (11 slices from 4 animals). Conversely, this proportion was 1.32 ± 0.19% on the ipsilateral side of the brain after the burn injury, with average counts of 68 out of 4953 (11 slices from 4 animals). The percentage of c-Fos-IR nuclei was 341.5 ± 59% in the contralateral side of the S1 cortex following the burn injury compared to the control (*p* = 0.003; Figure 1d). In contrast, the percentage on the ipsilateral side was 239.4 ± 45.7% (*p* = 0.049; Figure 1d).

The colocalization analysis of c-Fos and different subtypes of interneurons was conducted in the lower limb region of the S1 cortex, covering the entire cortical length (x/y dimensions: 500 μm × 800 μm). In this well-defined region of the S1 cortex in control mice, only 1.72 ± 0.33% of c-Fos-IR cells were colocalized with calbindin-positive neurons (Calb; 10 slices from 4 mice). Following burn injury, the colocalization percentage increased to 5.31 ± 0.75% in the contralateral S1 cortex and 10.07 ± 4.36% in the ipsilateral S1 cortex, with *p*-values of 0.012 and 0.008, respectively (11 slices from 4 animals; Figure 1e). In the S1 cortex of control mice, 2.34 ± 0.72% of c-Fos-IR cells were calretinin-positive neurons (CR; nine slices from four animals). Following a burn injury, this percentage increased significantly on the contralateral side to 8.2 ± 1.14% (*p* = 0.019; 11 slices from 4 animals), while the ipsilateral side remained significantly unchanged at 4.1 ± 1.22% (*p* = 0.59; Figure 1e). In control mice, 12.9 ± 2.7% of c-Fos-IR cells in the S1 cortex were identified as parvalbumin-positive neurons (Pvalb), and this proportion did not change following burn injury. In burn-injured mice, the contralateral S1 cortex displayed 11.8 ± 1.8% Pvalb-positive c-Fos-IR cells, while the ipsilateral cortex showed 9.5 ± 1.2% (six slices from two animals; Figure 1e). While Pvalb- and CR-expressing neurons were distributed throughout the entire S1 cortex, Calb-positive interneurons were predominantly located in the superficial layers, particularly in layers 2/3. Therefore, confocal images with a higher magnification of burn-injured mice were captured in layer 2/3 to illustrate the colocalization of c-Fos with different subtypes of interneurons (Figure 1f,g).

These findings indicate that burn injury profoundly influences the activity of various cortical neuron subpopulations, with a particular impact on calbindin- and calretinin-positive interneurons.

### 2.2. Burn Injury Is Accompanied by Moderate Transcriptomic Changes in the Primary Somatosensory Cortex as Early as One-Hour Post-Burn Injury

RNA-seq was performed to evaluate the impact of BI on pain-related gene expression profiles. We also investigated changes in gene expression in an inflammatory pain model induced by formalin application to assess whether there are differences in gene expression patterns between the two types of painful stimuli (Figure 2a). Samples were collected from the S1 cortex of wild-type mice that had been exposed to painful stimuli. Our RNA-seq analysis reveals significant differences at the transcriptomic level between the treated and untreated control groups. The corresponding findings are summarized in Figure 2 and the online Appendix A.

Using next-generation RNA-seq, a total of 13,339 counts or sequences were identified in each experimental group. Based on the criteria (−1 < logFC < 1), we identified a total of 1116 differentially expressed genes (DEGs) in the S1 cortex of BI mice, while 136 DEGs were detected in FA mice compared to the untreated control sample. Furthermore, we noted that only a small number of genes overlapped between the experimental groups. This comprehensive analysis, using a fold change (FC) of ≥2, reveals that 196 genes were upregulated in the mouse S1 cortex following burn injury and 92 genes after inflammatory pain (Figure 2b). Among these, 42 genes were elevated in both pain models, accounting for 21.4% of the upregulated genes in BI and 45.6% in inflammatory pain (Figure 2b). According to the criteria (−1 > logFC), 920 genes were downregulated in the BI samples, while 44 were downregulated in the FA samples (Figure 2b). Only 17 genes were negatively affected by both nociceptive stimuli, accounting for 1.8% of the BI group and 38.6% of the FA group (Figure 2b).

Heat maps illustrate the 15 most upregulated and downregulated genes in response to burn injury and formalin application. RNA-seq shows moderate changes in gene expression in response to painful stimuli compared to the control group (Figure 2c,d). After burn injury, the log2 fold changes in gene expression ranged from 2.69 to −3.42, while formalin treatment resulted in log2 fold changes ranging from 1.84 to −1.53 (Appendix A). Among the transcripts that were exclusively affected by burn injury, MT-ATP6 demonstrated the most substantial increase in expression, whereas DCC exhibited the most pronounced decrease (as detailed in Appendix A; Figure 2c,d). In contrast, PCP2 was the most significantly upregulated gene due to formalin treatment, and peripherin was the most downregulated gene affected exclusively by formalin injection (refer to Appendix A; Figure 2c,d). Among the genes with well-known functions that are similarly regulated by both noxious stimuli, MT-ATP8 exhibited the highest level of upregulation, as illustrated in Figure 2c and Appendix A. In contrast, PRSS56, which is a serine protease, displayed the greatest level of downregulation, regardless of the treatment applied (Figure 2d; Appendix A).

In conclusion, burn injury elicits moderate alterations in the transcriptomic profile of the S1 cortex as early as one hour post-injury. These changes differ significantly from those observed in the inflammatory pain model established by formalin treatment.

### 2.3. Differential Cortical Mechanisms in Acute Pain Processing Within the Primary Somatosensory Cortex Following Burn Injury and Inflammatory Pain

Heatmaps of gene expression reveal that both formalin application and burn injury significantly altered the gene expression profile in the S1 cortex (Figure 2). In the subsequent in silico experiments, we conducted GO enrichment and KEGG analyses to elucidate nervous system-specific GO terms associated with DEGs. The results of the GO analysis are shown in Figure 3, which includes the Gene Ontology biological process (GO_BP), cellular compartment (GO_CC), molecular function (GO_MF), and KEGG pathways. Using the criteria of −0.5 logFC to 0.5, 4619 DEGs were filtered in BI, while 988 DEGs were filtered in FA for GO analysis.

In the context of burn injury (Figure 3a), the GO_BP analysis identifies two enriched terms associated with upregulated DEGs (ATP metabolic process and gene expression) and eighteen enriched terms for downregulated DEGs. The GO_CC analysis reveals four enriched terms for upregulated DEGs and twenty-eight enriched terms for downregulated DEGs. Regarding the GO_MF analysis, only one enriched term was identified for upregulated DEGs, while nine enriched terms were identified for downregulated DEGs. In relation to burn injury, two terms were enriched among upregulated DEGs (i.e., retrograde endocannabinoid (REC) signaling and ribosome), and twenty-four terms were enriched among downregulated DEGs according to KEGG pathways analysis. Appendix A provides detailed bioinformatics data, including the false discovery rate (FDR), *p*-values, and the number of genes for each analyzed GO term and KEGG pathway.

In contrast to burn injury, the formalin-induced inflammatory pain model (Figure 3b) predominantly exhibited upregulated terms. The GO_BP analysis identifies nine enriched terms for upregulated DEGs, while the GO_CC analysis found 14 enriched terms for upregulated DEGs and three for downregulated DEGs. The GO_MF analysis reveals four enriched terms for upregulated DEGs and two for downregulated DEGs. In the KEGG pathway analysis, ten enriched terms were upregulated, such as purine metabolism and protein digestion and absorption, while seven terms were downregulated (Figure 3b). Appendix A includes comprehensive bioinformatics data, such as FDR, *p*-values, and the count of genes, for each analyzed GO term and KEGG pathway.

In conclusion, the GO enrichment analysis shows a significant difference in the expression of genes associated with specific GO terms between the two pain models. In the burn injury model, the majority of GO terms were downregulated, with only a limited number of terms being upregulated. Conversely, upregulation dominated over downregulation following formalin treatment. This observation suggests that the molecular mechanisms underlying the pain response in these two models are fundamentally distinct.

### 2.4. Differential Pain-Related Signaling in the S1 Cortex: Insights from KEGG Pathway Analysis Across Sensory Modalities

KEGG enrichment analysis was performed on the Cytoscape platform using the criteria −0.5 < logFC < 0.5 (Figure 4). Figure 4a provides a comprehensive list of BI-affected KEGG pathways linked to synaptic functions and neuronal morphology, including axon guidance, focal adhesion, and extracellular matrix (ECM)–receptor interaction. Notably, retrograde endocannabinoid signaling was significantly upregulated, as evidenced by its relatively high enrichment. Interestingly, DEGs associated with axon guidance were downregulated following BI (Figure 4a). Although fewer genes are involved in the serotonergic synapse, burn injury had a significant impact on serotonergic synaptic processes, as shown by its high −log_10_(FDR) value (Figure 4a). Figure 4b presents a comprehensive list of KEGG pathways associated with signaling processes in response to burn injury. Among these pathways, REC signaling is the most upregulated, as indicated by its high enrichment value.

Despite impacting fewer genes overall, burn injury significantly affected the TNF-, Notch-, and PPAR-signaling pathways, as demonstrated by their comparatively high −log_10_(FDR) values.

Formalin-induced inflammatory pain activates pathways rather than suppressing them (Figure 4c). When applied to the hind paw, formalin primarily upregulates metabolic pathways in the S1 cortex, including purine metabolism and protein digestion/absorption. This is supported by positive enrichment and high −log_10_(FDR) values (Figure 4c).

These findings indicate that burn injury induces unique gene expression patterns in the primary somatosensory cortex, distinct from those observed in the formalin-induced inflammatory pain model.

### 2.5. Impact of Burn Injury on the Components of the Retrograde Endocannabinoid Network in the S1 Cortex

A KEGG graph generated by Pathview was used to visualize the REC signaling pathway (04723 4/17/17, provided by Kanehisa Labs). This pathway involves the release of endocannabinoids, such as anandamide and 2-AG, from postsynaptic neurons. These endocannabinoids then travel retrogradely to presynaptic neurons, where they regulate neurotransmitter release and influence synaptic plasticity, resulting in both short-term and long-term changes [25,26]. By utilizing Pathview, we mapped our experimental RNA-seq data onto this pathway to illustrate the role of these components in the REC signaling relevant to the molecular mechanisms underlying burn injury. Figure 5 overviews the REC network influenced by burn injury.

Despite an overall increase in REC signaling, as shown in Figure 3 and Figure 4a,b, several key genes were downregulated following BI, as illustrated in Figure 5. Specifically, the cannabinoid receptor 1 (CB1R) gene, inward-rectifying potassium channels (referred to as GIRKs in Figure 5), and certain voltage-gated L-type calcium channels (noted as VGCCs in Figure 5) exhibited decreased expression. The observed results may be attributed to multiple factors, including the temporal dynamics and cell-type specificity of the components involved in REC signaling. However, exploring these details was beyond the scope of the current study. Further research is necessary to understand these factors and gain insight into the complex regulatory mechanisms that influence the pain response to burns.

Moreover, our analysis of RNA-seq data using Pathview reveals that synaptic vesicle cycling, crucial for releasing neurotransmitters from presynaptic terminals, was reduced (Figure 5). In contrast, the overall process of oxidative phosphorylation, vital for energy production, was found to be enhanced in the somatosensory cortex following burn injury (Figure 5).

### 2.6. Unlike CB1 Receptors, Ggamma14 Fluorescence Intensity in the S1 Cortex and the Number of Ggamma14-IR Cells in Layer 5 of the S1 Cortex Show Significant Alterations One Hour After Burn Injury

Peripheral burn injury can affect and overstimulate the REC network in the cortex (see Figure 3 and Figure 4). Despite its overall upregulation, specific key components, such as CB1Rs, exhibit decreased gene expression, while others, like the G protein subunit gamma 14 (Ggamma14), a member of the heterotrimeric G protein complex, show increased expression (Figure 5). To validate these observations at the protein level, we employed an immunofluorescent approach targeting both CB1R and Ggamma14.

Cannabinoid receptors are distributed throughout the pain circuits, extending from peripheral sensory nerve endings to the brain [27]. The CB1R is located in multiple layers of the rat somatosensory cortex, with notable expression in layers 2/3 and 5/6 [28]. These layers exhibit a high density of CB1R-positive fibers, indicating significant endocannabinoid signaling activity in the S1 cortex [28]. Consistent with this observation, we obtained similar results in mice (Figure 6a,b). The fluorescence intensity of the CB1R showed a slight reduction one hour after burn injury, decreasing from 1213.4 ± 70.6 to 911.8 ± 80.5 across the entire length of the contralateral S1 cortex; however, it did not reach statistically significant levels when compared to the control group (5–8 sections from 2 animals per group; Figure 6c). Appendix A shows that the fluorescence intensity of the CB1R did not vary significantly across different layers of the S1 cortex.

Ggamma14, encoded by the GNGT2 gene, acts as a part of the G protein complex that plays a role in cellular signaling [27,28]. While direct studies linking G gamma signaling to the endocannabinoid system are limited [29,30], its role in cellular signaling suggests potential involvement in this pathway. Further evidence supporting this hypothesis is provided by the KEGG Pathview findings (Figure 5). Representative confocal images of Ggamma14-IR in layers 2/3 and 5 from both a control mouse and a burn-injured mouse are displayed in Figure 6a,b. We observed a significant increase in the relative fluorescence intensity along the entire length of the contralateral S1 cortex after burn injury compared to the control group (*p* = 0.043; Figure 6d). Specifically, the Ggamma14 fluorescence intensity increased to 106.1 ± 19.7% on the contralateral side and decreased to 95.8 ± 27% on the ipsilateral side (3–5 sections from 2 animals per group; Figure 6d). Interestingly, a noticeable increase was detected in the number of Ggamma14-IR cells in layer 5 post-burn injury (Figure 6e). Layer 5, with its pyramidal neurons, plays a pivotal role in integrating sensory information and transmitting outputs to diverse cortical and subcortical regions. The percentage of Ggamma14-IR cells significantly increased to 202.5% ± 22 in layer 5 of the contralateral S1 cortex (*p* = 0.015), while it remained unchanged at 94.8% ± on the ipsilateral side (Figure 6e).

In summary, a decrease in CB1R mRNA was observed in the somatosensory cortex one hour post-burn injury. However, no significant difference was detected at the protein level throughout the observation period. This mismatch between mRNA and protein levels at the one-hour mark is likely due to the CB1 receptor’s stable half-life of around 24 h, allowing it to sustain functionality over time. In the contralateral S1 cortex, there was a noticeable increase in Ggamma14 fluorescence intensity throughout its entire length, which coincided with a substantial rise in Ggamma14-IR cell numbers, especially in layer 5. This highlights the potential involvement of pyramidal neurons in layer 5 in the functional remodeling triggered by burn injury, likely mediated through G-protein signaling pathways.

## 3. Discussion

Burn injury is a chronic condition marked by increased sensitivity and spontaneous pain [1,2,3,4]. Its management is often inadequate due to a limited understanding of the underlying mechanisms. Like other chronic pain conditions [5,6,13,15,16,17,18], BI can cause lasting changes in neural activity in specific brain regions, altering sensory perception and pain experiences. Thus, understanding the molecular mechanisms of pain processing in burn injury is crucial. We recently found that pain models from different origins, like burn injury and formalin application to the hind paw, lead to distinct gene expression patterns in the mouse spinal cord [20]. We hypothesized that burn injury-induced pain also triggers transcriptomic alterations in the S1 cortex, which may account for the subsequent pain-related behaviors observed afterward.

To elucidate the pathogenesis and neurobiological mechanisms underlying burn injury-induced pain, we conducted transcriptome-wide RNA-seq. This was followed by a comprehensive bioinformatic analysis to interpret the extensive data generated. We compared the changes in the gene expression pattern caused by burn injury to those induced by formalin-induced inflammatory pain models. Our hypothesis was that pain models originating from various etiologies induce distinct transcriptomic alterations in the S1 cortex, similar to those observed in the spinal cord [20]. We identified DEGs and clustered them into signaling pathways and functional gene networks to better understand the molecular mechanisms involved in pain processing. KEGG pathway analysis reveals the involvement of REC signaling in the modulation of burning pain within the S1 cortex of mice. Additionally, two components of REC signaling that had exhibited transcriptional changes following burn injury were validated at the protein level using immunofluorescent staining (the CB1 receptor and Ggamma14).

RNA-seq identified 1116 and 136 DEGs in the S1 cortex of burn-injured and formalin-treated mice, respectively, using the criteria of −1 < logFC < 1 compared to untreated mice. The number of downregulated genes following BI was six times higher than that of upregulated genes (920 versus 196). This significant gene suppression may be attributed to two main factors: first, epigenetic modifications, including DNA methylation [11,20]; and second, the activation of stress responses, which tend to prioritize certain pathways (e.g., stress responses) over normal cellular functions. GO enrichment analysis reveals that BI significantly altered GO terms related to neuronal functions, including synaptic plasticity, excitability, and axon guidance. In contrast, formalin treatment notably impacted cell cycle-related processes and protein digestion and absorption, as well as the metabolism of phospholipids and carbohydrates. This underscores that different painful modalities induce distinct molecular and biochemical processes, resulting in unique transcriptional patterns in the cortex depending on the modality.

In the S1 cortex of wild-type mice following burn injury, significant upregulation was observed in genes linked to oxidative phosphorylation (MT-ATP6), glycosphingolipid biosynthesis (A4GALT), and epithelial cell differentiation (e.g., LRG1, TAGLN, and RBM46). The MT-ATP6 gene plays a vital role in mitochondrial oxidative phosphorylation and the production of ATP. Notably, MT-ATP6 expression is increased in spinal cord samples from both pain models [20]. However, in the S1 cortex, it is overexpressed exclusively after burn injury, likely due to heightened mitochondrial stress. In contrast, the expression of MT-ATP6 decreased in nociceptive dorsal root ganglion (DRG) neurons following sciatic nerve injury [31]. Burn exposure leads to the downregulation of several key genes, including those involved in semaphorin signaling (SEMA5A and PLXNA4), axon guidance (DCC and ERBB4), cytoskeleton remodeling (PLXNA4), and pathways related to synaptic plasticity and neurotransmission (e.g., CBLN1, GRIN2A, GRIK3, and BSN). These findings align with Hozumi’s study, which demonstrated that chronic pain alters neuronal structure in the spinal cord by influencing genes involved in axon and dendrite growth [32]. Interestingly, DCC is highly expressed in the CNS and is associated with the basal lamina of epithelial cells [33]. Elevated levels of DCC may, therefore, enhance the structural integrity of the blood–brain barrier by promoting the proper organization and maintenance of endothelial cells and their tight junctions. Chronic pain is associated with anatomical changes, such as spine remodeling and synaptic reorganization in the spinal cord [34]. In line with the KEGG Pathview shown in Figure 5, our findings suggest that burn injury downregulated genes critical for synaptic plasticity and the regulation of neuronal activity, including the glutamate receptor 7-encoding GRIK3 and the NMDA receptor subunit 2A-encoding GRIN2A. Numerous studies have established that the BSN gene (Bassoon), which encodes a scaffold protein in the presynaptic cytomatrix, is crucial for presynaptic function [35,36]. A decrease in BSN levels may indicate reduced synaptic activity and the potential downregulation of synaptic transmission. Again, the KEGG Pathview analysis derived from our current RNA-seq data supports this assumption. ERBB4 transcription is crucial for axon guidance and the activation of the MAPK1/ERK2 and MAPK3/ERK1 signaling pathways, which are essential for cell proliferation, neurite outgrowth, and heat sensation in the spinal cord [37,38,39]. Consequently, reduced ERBB4 transcription levels are associated with dysfunction in all these activities. Overall, the downregulation of these genes resulting from burn injury may cause impaired synaptic function, decreased neural plasticity, and potential deficits in sensory processing.

In total, 136 DEGs were detected after formalin treatment in wild-type mice compared to the untreated sample, of which 92 were upregulated and 44 were downregulated. Significant enrichment of upregulated genes was observed in G protein-mediated cell signaling (e.g., PCP2, ARHGEF33, and NPR1), transcriptional and translational regulation (e.g., IKZF1, EIF4EBP3, ATOH8, and CYREN), phospholipid homeostasis (e.g., PLA2G4A and CYP39A1), and glycosaminoglycan (GAG) biosynthesis (CHST14). Among the top 15 downregulated genes, several are implicated in glycosaminoglycan metabolism (CHST5) and fatty acid metabolism (NUDT7), while others are associated with the cAMP signaling pathway (KCTD14) or ERK/MAPK signaling (e.g., PDGFC and TIRAP). The variation in gene expression related to cholesterol and glycosaminoglycan synthesis, with some genes being upregulated (e.g., PLA2G4A, CYP39A1, and CHST14) and others downregulated (e.g., CHST5 and NUDT7), can be attributed to cell type-dependent metabolic shifts that occur during inflammation. Interestingly, the expression of NUDT7 decreased following formalin injection but increased after BI.

By comparing the two pain models, we identified 59 DEGs in the S1 cortex that responded similarly to noxious treatments. Of these genes, 42 were upregulated, while 17 were downregulated. We observed elevated levels of energy-related metabolic pathway genes (e.g., MT-ATP8 and MT-CO2), transcriptional regulatory genes (ZFP459 and 764), and stress-responsive genes (e.g., HBA-A2) in pain models, regardless of their origin. Our findings are consistent with previous research indicating that elevated levels of HBA genes, especially HBA-A2, play a role in the cellular response to social stress in the prefrontal cortex [40] and in response to spinal cord injury [41]. Additionally, HBA2 has been identified in various brain regions, indicating potential roles that extend beyond its known function in oxygen transport (https://www.informatics.jax.org/marker/MGI:96016; accessed on 19 January 2025). Genes associated with oxidative phosphorylation and mitochondrial homeostasis, specifically components of cytochrome c oxidase (MT-CO2) and the mitochondrial proton-transporting ATP synthase complex (MT-ATP8), exhibited upregulation in both nociceptive conditions. Consistent with this observation, a significant increase in the levels of mt-Co2 and mt-Atp6 has been demonstrated in the tibial nerve injury model, as measured by the mitochondria-specific marker Tomm20 [13]. In contrast to our findings, mitochondrial dysfunction has been shown to significantly contribute to the pathophysiology of neuropathic pain [42]. This dysfunction is characterized by the repression of ATP synthase and a reduction in electron transport chain activity by day 7 following the partial sciatic nerve ligation model. The discrepancies between the two studies may be attributed to the different pain models used and the timing of sample collection post-injury (7 days versus one hour). Both treatments led to a downregulation of genes associated with G-protein-coupled receptors (HTR2C and BDKRB2), ion channels (P2RX3 and KCNJ2), and ECM remodeling (GLB1L and THBS4). HTR2C is the gene responsible for encoding the 5-HT2C receptor, a subtype of serotonin receptor that plays a critical role in the regulation of energy homeostasis [43] as well as in the development of neuropathic pain-related behaviors [44]. BDKRB2 is the gene that encodes the bradykinin receptor B2. Bradykinin, a pro-inflammatory peptide, interacts with its receptors to activate pain pathways, which can result in inflammation and neuropathic conditions, as previously reported in models of brachial plexus avulsion [45]. Nociceptive stimuli negatively affected the P2X3 receptor for ATP, which is essential for sensory perception [46], and the KCNJ2 potassium channel, which is critical for neuronal excitability [47]. Glycosaminoglycans (GAGs) generally elicit an anti-inflammatory response in pain conditions, including osteoarthritis [48]. However, genes involved in GAG metabolism, such as GLB1L, were downregulated in the acute pain models studied. Thrombospondin-4 (TSP4), which is encoded by the THBS4 gene, is a key regulator of ECM organization and cell-to-cell and cell-to-matrix interactions [49]. It has been reported that TSP4 expression is elevated in the spinal cord following peripheral nerve injuries, contributing to the development of neuropathic, bone cancer, and inflammatory pain [49,50,51,52]. In contrast, we found that THBS4 was downregulated in the S1 cortex. Consequently, we hypothesize that reducing TSP4 levels may attenuate this hyperexcitability, potentially acting as a protective mechanism to prevent the transition from acute to chronic pain at the onset of injury [49,50,51,52].

Through GO enrichment analysis of the RNA-seq data, we identified several gene clusters involved in key biological processes. Following burn injury, the upregulated genes were predominantly enriched in mitochondrial functions and processes related to gene expression, including ATP metabolism and ribosome activity. In contrast, many downregulated DEGs were associated with axon guidance, synaptic plasticity, and neurotransmission. In the context of burn injury, KEGG pathway analysis identified significant enrichment in several biological processes, including synaptic function, axon guidance, serotonergic synapses, and various signaling pathways, such as TNF, Notch, and PPAR. Furthermore, the analysis emphasized the significance of the retrograde endocannabinoid signaling pathway. In contrast, formalin application leads to the enrichment of genes involved in protein digestion and absorption, phospholipid and carbohydrate metabolism, and processes associated with the cell cycle.

REC signaling plays a vital role in endogenous pain control [53,54,55]. Nonetheless, the specific involvement of its components in the cortical processing of pain associated with burn injury remains unexplored. When neurons in the cortex are activated by painful stimuli, they release endocannabinoids. These endocannabinoids travel backward across synapses to bind to presynaptic cannabinoid receptors. This binding inhibits the release of both inhibitory and excitatory neurotransmitters, which reduces overall neuronal excitability and alters pain perception [53,54]. Thus, CB1R activation reduces chronic pain and modulates sensory processing [56,57]. Cortical CB1R downregulation can have several functional consequences in the context of pain processing, including reduced pain modulation and altered neurotransmitter release [58,59]. When CB1Rs are downregulated, their ability to inhibit adenylate cyclase and reduce cAMP levels diminishes, leading to decreased inhibition and resulting in heightened pain sensitivity [60]. In addition to its well-known presence in the plasma membrane, the CB1R is also found in mitochondrial membranes, suggesting a direct role in regulating mitochondrial function [61,62]. In this location, it reduces mitochondrial respiration and contributes to “depolarization-induced suppression of inhibition” by preventing the release of the inhibitory neurotransmitter GABA from inhibitory terminals [63,64]. Despite the repression of several key proteins involved in REC signaling following burn injury (e.g., CB1R and CACNA1C), KEGG pathway analysis reveals an overall upregulation of the signaling process. This upregulation can be attributed to the increased expression of components involved in oxidative phosphorylation, highlighting the significant interrelationship between these two processes. Activation of mitochondrial CB1Rs can reduce mitochondrial respiration, impacting energy production and cellular metabolism [64,65]. Consequently, the downregulation of CB1Rs can lead to increased mitochondrial respiration and ATP production, improving cellular metabolism and reducing oxidative stress.

The potential interactions between CB1Rs and Ggamma14 in signaling pathways can be quite intricate. The Gγ14 subunit, part of the Gβγ complex, plays a role in enhancing the specificity and efficiency of CB1R signaling. When CB1R is activated, it interacts with a heterotrimeric G-protein complex made up of three subunits: alpha (Gα), beta (Gβ), and gamma (Gγ). This interaction leads to the initiation of various alternative downstream pathways, such as the inhibition of adenylyl cyclase and activation of mitogen-activated protein kinases (MAPKs), resulting in changes in gene expression and the regulation of ion channels, including the inwardly rectifying potassium channel encoded by GIRK (Figure 5) [66,67]. Therefore, elevated levels of Ggamma14, along with the downregulation of CB1Rs, may indicate a compensatory mechanism in pain signaling pathways. This interplay highlights the complexity of CB1R signaling and the potential role of Ggamma14 in balancing pain control homeostasis in the S1 cortex.

To the best of our knowledge, this study represents the first comprehensive analysis of gene expression changes and the associated key signaling pathways in the primary somatosensory cortex of a mouse model subjected to burn injury utilizing whole transcriptome bulk RNA-seq. However, several limitations inherent to this study must be recognized. Firstly, tissue collection for RNA-seq was performed 1 h after the induction of nociceptive stimuli, representing the acute phase of pain models. However, due to animal welfare considerations, it was not feasible to examine changes in gene expression at multiple time points. This ethical limitation restricts our ability to gain a comprehensive understanding of the shift of gene expression profiles and the associated dynamics of energy demand following BI. Sampling at different intervals would likely produce varying expression patterns. Secondly, neurons and glial cells likely undergo activity-dependent changes in response to tissue injury and pain, influenced by epigenetic tagging [68,69]. This study did not investigate these changes. Future research should employ single nucleus RNA-seq for the detailed genetic profiling of specific cell subpopulations. Thirdly, previous studies have suggested that the ipsilateral and contralateral brain regions may have different roles in pain processing [17,70]. However, in our RNA-seq experiments, we did not analyze the two sides of the brain separately; both hind legs were exposed to nociceptive stimuli. As a result, both sides of the brain were considered contralateral to the stimulus. Additionally, we did not investigate the specific changes in the transcriptome resulting from pain stimuli at the level of cortical columns and layers. Therefore, further research is needed to explore any layer-specific differences. Fourthly, a single bulk RNA-seq run was conducted using pooled biological samples from 12 to 13 animals per group rather than employing multiple replicates. Ultimately, while RNA-seq is an effective preliminary screening method, it is essential to conduct further direct experiments to determine whether the transcriptional changes observed in the S1 cortex are linked to changes in neuronal function.

This study enhances our understanding of the molecular mechanisms associated with burn injury. It demonstrates that different painful stimuli activate distinct transcriptional programs in the S1 cortex depending on the sensory modality, with minimal overlap between pain models. The identified genes and molecular pathways are pivotal in determining the functional outcomes of various pain conditions and possess significant clinical relevance. They also offer potential targets for further experimental hypotheses and therapeutic interventions.

## 4. Materials and Methods

### 4.1. Animals, Ethical Considerations, and Study Design

The experiments were conducted on adult male mice, with approval from the Animal Care and Protection Committee at the University of Debrecen (Approval No.: 15-1/2023/DEMÁB). All procedures adhered to the European Community Council Directives and the IASP Guidelines. The animals were housed individually in a temperature-controlled colony room, maintained on a 12 h light/dark cycle, and provided with food and water ad libitum.

A total of 50 wild-type CD1 mice were utilized in this study. These animals were obtained from Charles River Laboratories (Wilmington, MA, USA). For RNA-seq, 38 mice were divided into three groups: a control group (n = 13), a burns group (n = 13), and a group treated with formalin (n = 12). Twelve animals were utilized for immunostaining studies (six for burn injury and six for control). Painful stimuli were applied to the hind legs of the mice. In control cases, the animals were left untreated.

### 4.2. Establishment of Burn Injury Model

A comprehensive methodology for establishing a non-severe burn injury model is detailed in references [71,72]. Briefly, under deep anesthesia induced by sodium pentobarbital (Release, 50 mg/kg intraperitoneal; WDT, Garbsten, Germany; #085093), mice were subjected to a full-thickness burn injury, affecting the entire hind leg up to the knee joint, by immersing them in 60 °C hot water for two minutes. This represents a burn of about 5% of the total body surface area (TBSA), consistent with a non-severe burn model [73]. Both hind legs of thirteen mice were simultaneously exposed to burns for subsequent transcriptome analysis. Only the left leg of 6 mice was subjected to thermal stimulus for subsequent immunostaining studies.

### 4.3. Establishment of Formalin-Induced Inflammatory Pain Model

In the formalin-induced inflammatory pain model, 25 μL of a 5% formaldehyde solution was injected intraplantarly under the plantar pads of both hind legs [74] while the mice were under deep anesthesia (50 mg/kg i.p. sodium pentobarbital). This experiment involved 12 CD1 wild-type mice, and it was used for transcriptome analysis.

### 4.4. Immunofluorescent STAINING

Six control and six burn-injured wild-type mice were utilized for this study experiment. After a one-hour survival period, the experiment was terminated without the animals regaining consciousness, and they were perfused transcardially with a 4% paraformaldehyde (PFA) solution under deep sodium pentobarbital anesthesia. The bilateral S1 cortical region was dissected, post-fixed in 4% paraformaldehyde (PFA) for 3 h, and sectioned at 100 µm thickness using a vibrating blade vibratome (VT 1000S; Leica Biosystems, Wetzlar, Germany). Immunofluorescent staining was outlined as described in our previous studies [71,75], with the following modification: antigen retrieval was performed in a sodium–citrate buffer (pH 6.0) containing 0.05% Tween 20 at 95 °C for 30 min. Briefly, the primary antibody was incubated at 4 °C for two days, followed by an overnight incubation with the secondary antibody. The primary antibodies used in this study are detailed in Table 1.

Species-specific secondary antibodies conjugated to Alexa Fluor-488, 555, and 647 (Invitrogen, Waltham, MA, USA) were utilized. To identify cell nuclei and layer boundaries, 4′,6-diamidino-2-phenylindole (DAPI) staining was applied based on its density. The sections were subsequently mounted in Hydromount (National Diagnostics, Brandon, FL, USA) and imaged using Olympus FV3000 confocal systems (Tokyo, Japan).

### 4.5. Total RNA Isolation and RNA-Seq Analysis

After an hour of survival, the experiment was terminated by perfusing the mice with ice-cold sterile saline without the animals regaining consciousness. The cortical regions associated with the S1 cortex were immediately excised and placed into TRIzol™ Reagent (Thermo Fisher Scientific Inc., Waltham, MA, USA) for RNA-seq analysis, as detailed in [20]. The RNA concentration was quantified using a NanoDrop spectrophotometer. RNA samples from individual animals were pooled in equal amounts into three treatment-specific groups. The quality of the pooled RNA was then assessed using an Agilent BioAnalyzer and the Eukaryotic Total RNA Nano Kit (Agilent Technologies, Santa Clara, CA, USA), following the manufacturer’s protocol.

RNA-seq was performed, as described previously by [20]. Briefly, samples with an RNA integrity number (RIN) above 8 were selected for library preparation using the NEBNext^®^ Ultra II RNA Sample Preparation Kit (New England BioLabs, Ipswich, MA, USA). Sequencing was conducted on the Illumina NextSeq 500 instrument (Illumina, San Diego, CA, USA) with single-end 75-cycle runs. Raw .bcl files were converted to fastq files and demultiplexed using Illumina’s bcl2fastq software v1.8.4. The data were aligned and normalized with the STAR aligner (version 2.7.11; [77]) and mapped to the mouse genome (Ensembl, GRCm39). Differentially expressed genes were identified using edgeR (version 3.18). Gene expression data were visualized with the R packages VennDiagram and ggplot2 (version 4.3.3), along with SRPLOT [78]. Gene pathway analysis was conducted using Cytoscape (version 3.10.1; [79]) with the built-in String and EnrichmentMap apps. Aligned sequencing data have been deposited into the NCBI SRA database under accession number PRJNA1137110.

### 4.6. Data Analysis

For Gene Ontology KEGG pathway analysis, the false discovery rate (FDR) was used to indicate the correlation strength between the identified genes in our dataset and all the genes described in a given pathway, employing the built-in algorithm of the EnrichmentMap app in Cytoscape. In a KEGG bubble chart, the enrichment was calculated with the following formula: E = (#up − #down)/√#total genes, where E is the enrichment, #up is the number of upregulated genes, #down is the number of the downregulated genes, and #total is the number of all genes found in the particular KEGG pathway. It is used to signify the relative importance or significance of a pathway within the context of the analyzed data. All data obtained from immunostaining are presented as the mean ± standard error of the mean (SEM). Statistical analyses utilized Past4 software v1.0.0.0, employing a non-parametric Mann–Whitney U-test.

## 5. Conclusions

We investigated transcriptomic alterations in the primary somatosensory cortex using bulk RNA sequencing across two pain models: a non-severe full-thickness burn injury model and a formalin-induced inflammatory pain model. This cortical area has not been thoroughly studied in relation to burn injuries, and our findings offer significant insights. Gene Ontology and KEGG enrichment analyses reveal that burn injury upregulates genes associated with mitochondrial functions and gene expression while downregulating those related to axon guidance and neurotransmission. Conversely, formalin treatment primarily affected metabolic processes, including protein digestion and purine metabolism. These findings suggest that transcriptomic changes in the primary somatosensory cortex depend on the type of sensory modality.

## Figures and Tables

**Figure 1 ijms-26-03538-f001:**
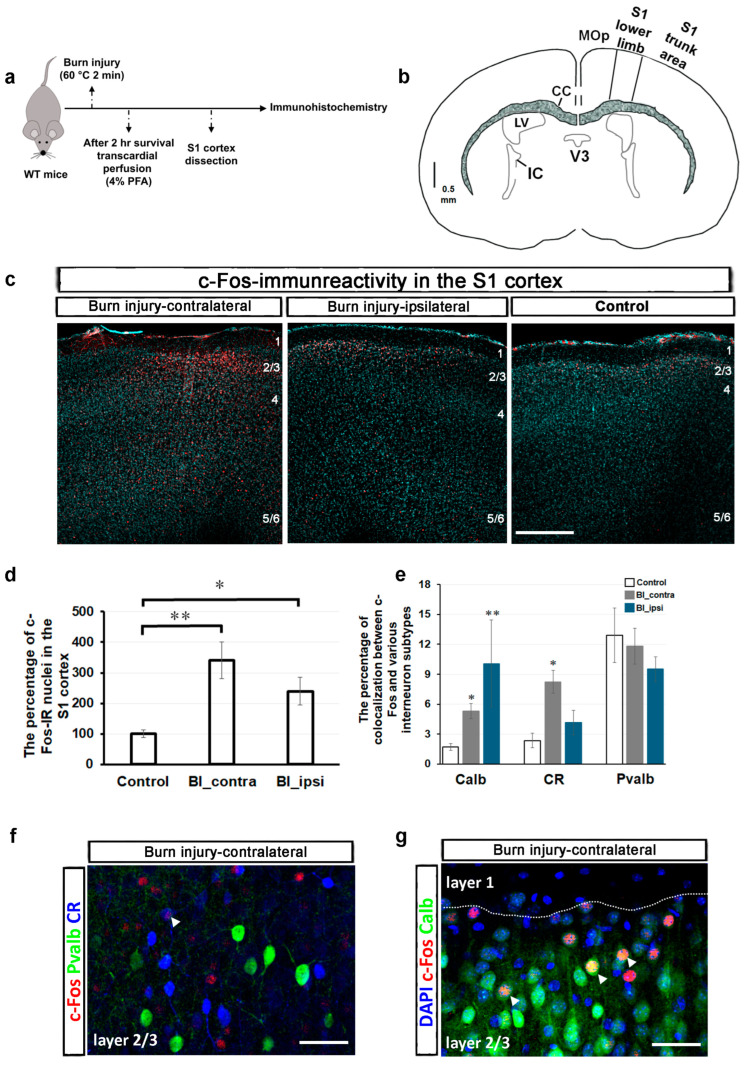
The impact of burn injury on cortical neuronal activity: c-Fos immunoreactivity (IR) analysis. (**a**) Schematic representation illustrating the time scale of this experiment. (**b**) Schematic drawing of a coronal section of the mouse brain representing the lower limb area of the S1 cortex. (**c**) Representative immunostaining with an antibody against c-Fos (red) was obtained from three optical sections of a coronal slice of a wild-type mouse brain following burn injury. DAPI counterstained cell nuclei for turquoise. After the burn injury, the number of cells exhibiting c-Fos-IR increased in the lower limb area of the S1 cortex, irrespective of the sides. The term “burn injury-contralateral” specifically refers to the right hemisphere of the brain, which is opposite the burn injury site. Scale bar, 220 μm. (**d**) Quantitative analysis of the percentage of c-Fos-IR nuclei in the lower limb area of the S1 cortex following burn injury. The data are presented as mean ± SEM (10–11 slices from 4 animals per group). An asterisk denotes a *p*-value < 0.05, while a double asterisk indicates a *p*-value < 0.01, vs. control. (**e**) Quantitative analysis of the percentage of colocalization between c-Fos and various interneuron subtypes in the S1 cortex following burn injury. The data are presented as mean ± SEM (5–11 slices from 2–4 animals per group). * signifies *p* < 0.05, while ** indicates *p* < 0.01, vs. control. (**f**) Representative triple immunostaining with antibodies against c-Fos, parvalbumin (Pvalb), and calretinin (CR) from layer 2/3 of the contralateral-sided S1 cortex following burn injury. Arrowhead shows a CR-positive interneuron that expresses c-Fos. Scale bar, 30 μm. (**g**) Representative triple immunostaining with antibodies against c-Fos, calbindin (Calb), and a nuclear marker, DAPI, from layer 2/3 of the contralateral-sided S1 cortex following burn injury. The dotted line indicates the boundary between layer 1 and layer 2/3. Arrowheads show colocalizations of c-Fos and Calb. Scale bar, 30 μm. Abbreviations: WT, wild type; PFA, paraformaldehyde; S1, primary somatosensory cortex; MOp, primary motor cortex; V3, third ventricle; IC, internal capsule; LV, lateral ventricle; CC, corpus callosum.

**Figure 2 ijms-26-03538-f002:**
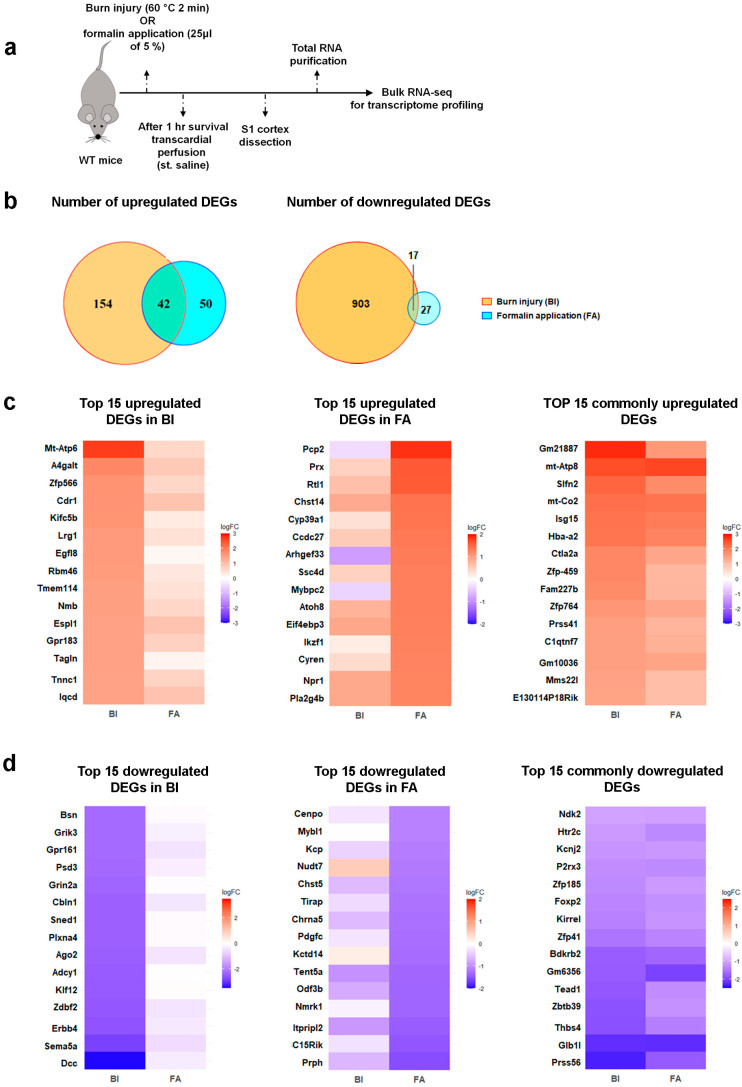
Distinct transcriptomic profiles of the S1 cortex in two pain models. RNA-seq analysis reveals limited overlap in DEGs between burn injury and inflammatory pain models. (**a**) Schematic representation illustrating the time scale of this experiment. (**b**) The Venn diagram analysis of DEGs in the S1 cortex illustrates the overlap of DEGs (logFC between −1 and 1, relative to untreated samples) between the BI model and the formalin-induced inflammatory pain model. The intersections represent DEGs commonly dysregulated in response to these painful conditions. (**c**) The heatmap of RNA-seq data illustrates the top 15 upregulated DEGs in the S1 cortex following burn injury (BI, shown in the left panel) or formalin application (FA, seen in the middle panel). The panel on the right lists 15 commonly upregulated DEGs across the pain models (out of the 42 genes shown in the intersection of the above Venn diagram). In each panel, color-coded expression values of identical genes are displayed side by side to compare the two pain models. Blue indicates downregulated genes, while warm colors represent upregulated ones. (**d**) This heatmap of RNA-seq data illustrates the top 15 downregulated DEGs in the S1 cortex following burn injury (BI, on the left panel) or formalin application (FA, in the middle panel). The right panel lists 15 (out of 17) commonly downregulated DEGs across the pain models. In each panel, the expression values of identical genes are color-coded and displayed side by side to compare the two pain models. Blue indicates downregulated genes, while warm colors indicate upregulated ones. Appendix A detail the molecular functions and biological processes related to the top 15 up- and downregulated DEGs.

**Figure 3 ijms-26-03538-f003:**
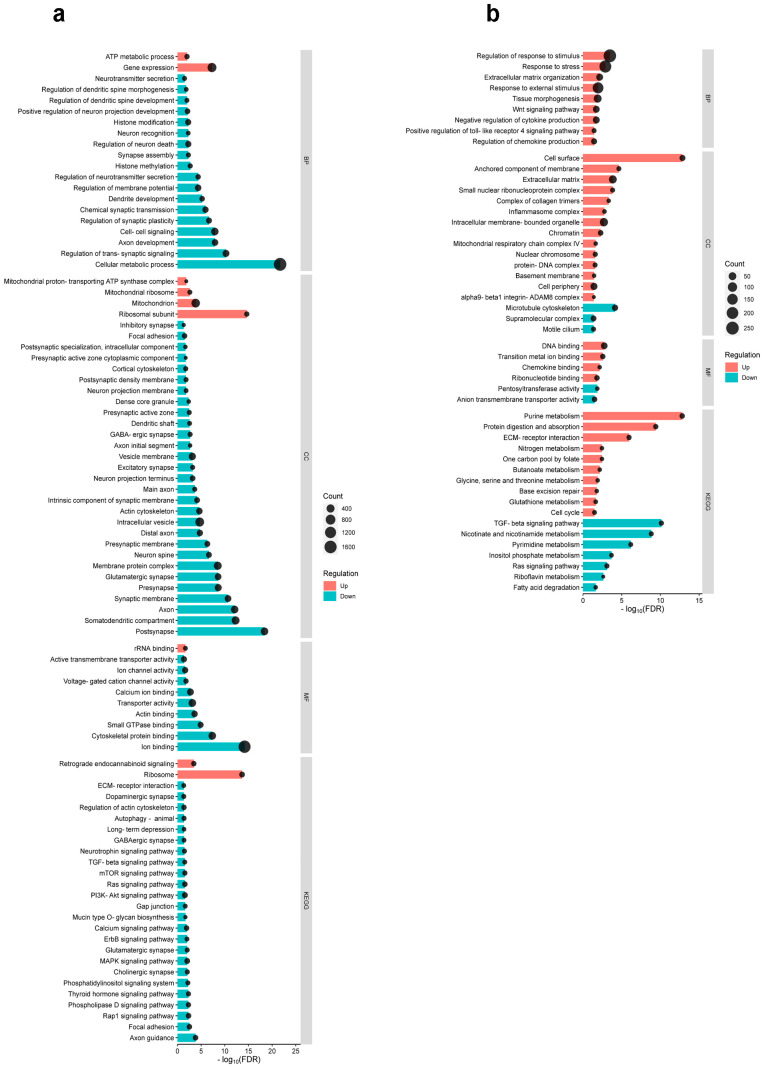
GO enrichment analysis reveals distinct molecular signaling pathways activated by thermal injury and formalin-induced inflammatory pain in the S1 cortex. (**a**) The bar–dot plot illustrates the results of the GO terms and KEGG pathway enrichment analysis of the DEGs in the S1 cortex in response to BI compared to the control. (**b**) The bar–dot plot illustrates the results of the GO terms and KEGG pathway enrichment analysis of the DEGs in the S1 cortex in response to formalin-induced inflammation compared to the control. The vertical axis lists the terms, while the horizontal axis displays the transformed false discovery rate (−log_10_(FDR)). Upregulated terms are depicted in red and downregulated terms in blue based on the ratio of the numbers of up- and downregulated genes in each entry. Line lengths indicate the significance level, and dots represent the number of genes. Abbreviations: BP, Gene Ontology biological process; CC, Gene Ontology cellular compartment; MF, Gene Ontology molecular function; KEGG, Kyoto Encyclopedia of Genes and Genomes.

**Figure 4 ijms-26-03538-f004:**
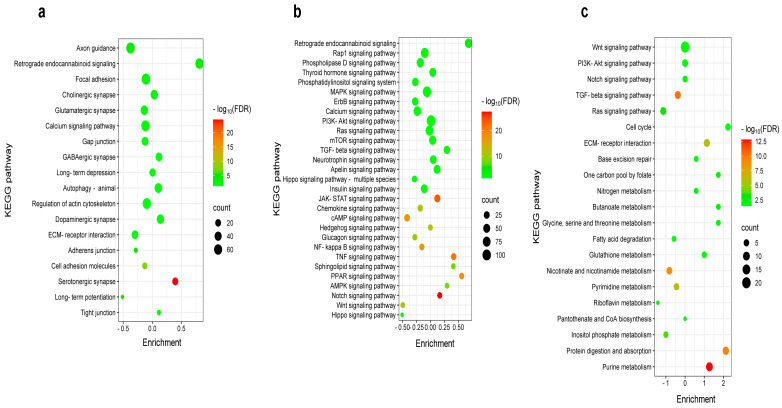
KEGG pathway analysis identifies distinct pain-related molecular programs in the S1 cortex based on sensory modality. (**a**) Bubble chart illustrating KEGG pathway enrichment for DEGs related to synaptic and structural functions in burn injury compared to the control group. (**b**) Bubble chart depicting KEGG pathway enrichment for DEGs related to signaling processes in burn injury compared to the control group. (**c**) Bubble chart illustrating KEGG pathway enrichment for DEGs related to signaling and metabolic processes in formalin treatment compared to the control group. The vertical axis represents KEGG pathway entries, while the horizontal axis shows the ratio of DEGs enriched in each pathway to the total number of genes annotated in that pathway (filtered by a log fold change criterion of −0.5 < log_2_FC > 0.5). The size of each bubble corresponds to the number of genes associated with a given pathway. The color gradient represents the false discovery rate (FDR) on a log_10_ scale, indicating statistical significance. The enrichment scale quantitatively expresses the proportion of genes in the given KEGG pathway that are up- or downregulated.

**Figure 5 ijms-26-03538-f005:**
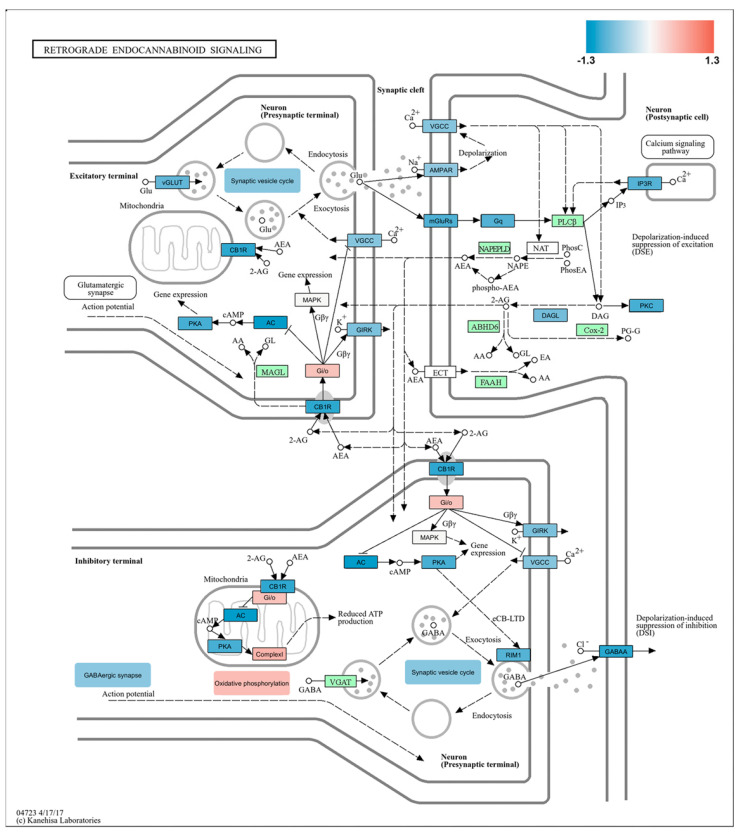
Identification of key players in burn injury-related REC network. KEGG Pathview analysis of the molecular signature of the REC signaling pathway in burn injury compared to control. This analysis illustrates the pre- and postsynaptic components of REC signaling at both excitatory and inhibitory synapses. The KEGG pathway analysis indicates that burn injury leads to the upregulation or downregulation of multiple components in the REC pathway. Each box represents a certain gene family or a single gene, with color indicating the log_2_-based fold change. Gene expression levels are shown as higher (red), unchanged (green), or lower (blue) in burn-injured samples compared to non-treated controls. The synaptic vesicle cycle shows downregulation in presynaptic terminals, regardless of the functional nature of the synapse. On the other hand, oxidative phosphorylation was typically upregulated in the axon terminals in response to burn injury. Abbreviations: CB1R, cannabinoid receptor type 1; VGCCs, voltage-gated calcium channels; GIRKs, inwardly rectifying potassium channels.

**Figure 6 ijms-26-03538-f006:**
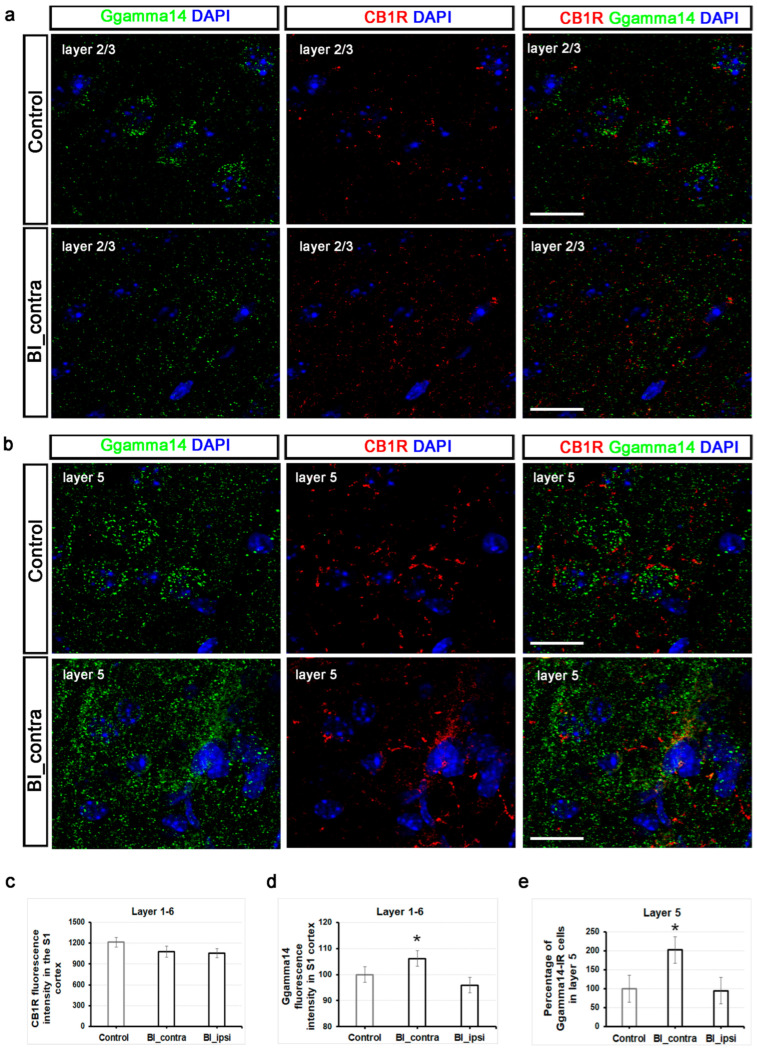
Validation of RNA-seq datasets using immunofluorescence: increased Ggamma14 fluorescence intensity and a higher number of Ggamma14-IR cells in the S1 cortex post-burn injury. (**a**) Representative triple immunofluorescent labeling of the S1 cortex in a control mouse (top) and a mouse subjected to burn injury (bottom) are shown, utilizing antibodies against the CB1R (red), Ggamma14 (green), and DAPI (blue) in layer 2/3. Scale bar: 10 µm. (**b**) Representative triple immunofluorescent labeling of the S1 cortex in a control mouse (top) and a mouse subjected to burn injury (bottom) are shown, utilizing antibodies against the CB1R (red), Ggamma14 (green), and DAPI (blue) in layer 5. Scale bar: 10 µm. (**c**) Quantitative analysis of CB1R fluorescence intensity throughout the entire thickness of the contralateral S1 cortex, relative to the control. The data are presented as mean ± SEM (n = 5–8 sections from 2 animals per group). (**d**) Quantitative analysis of fluorescence intensity for Ggamma14-IR throughout the entire thickness of the S1 cortex was normalized to the control. The data are presented as mean ± SEM (n = 3–5 sections from 2 animals per group). An asterisk (*) indicates *p* < 0.05 vs. controls. (**e**) Quantitative analysis of the percentage of Ggamma14-IR cells in layer 5 of the S1 cortex, relative to the control. The data are presented as mean ± SEM (n = 3–5 sections from 2 animals per group). An asterisk (*) indicates *p* < 0.05 vs. controls.

**Table 1 ijms-26-03538-t001:** List of the primary antibodies used in the study, including their suppliers, catalog numbers, and dilution factors.

Name	Species	Dilution	Supplier	Cat. No.
* c-Fos (K-25)	rabbit polyclonal IgG	1:1000	Santa Cruz Biotechnology, Inc., Dallas, TX, USA	sc-253
^€^ CB1 receptor	goat polyclonal IgG	1:1000	Frontiers Institute Co., Ltd., Ishikari, Japan	CB1-Go-Af450
Ggamma14	rabbit polyclonal IgG	1:250	Novus Biologicals, Centennial, CO, USA	NBP2-58646
calbindin D28k	guinea pig polyclonal IgG	1:250	Synaptic Systems GmbH.; Göttingen, Germany	214004
calretinin	guinea pig polyclonal IgG	1:500	Swant; Burgdorf, Switzerland	CRGp7
parvalbumin	mouse monoclonal IgG1	1:1000	Swant	PV-235

* c-Fos (K-25) has been discontinued and replaced by c-Fos (E-8): sc-166940. ^€^ see in [76].

## Data Availability

All data relevant to the study are included in the article or uploaded as online Appendix A.

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
