# Peer review of "The Burning Pain Transcriptome in the Mouse Primary Somatosensory Cortex"

_ijms, 2025, doi:10.3390/ijms26083538_

Round 1
Reviewer 1 Report
Comments and Suggestions for Authors
The study examines transcriptomic changes in the primary somatosensory cortex (S1) of mice following burn injury and formalin-induced inflammatory pain. RNA sequencing identified distinct gene expression profiles between the two pain models, with burn injury showing significant downregulation of genes related to synaptic plasticity and neurotransmission, while formalin treatment mainly affected metabolic processes. Gene Ontology and KEGG pathway analyses revealed that burn injury activated retrograde endocannabinoid signaling and mitochondrial functions, whereas formalin treatment influenced protein digestion and purine metabolism. The findings highlight the unique molecular mechanisms involved in pain processing for different pain types.
Minor comments:
Figure 1d Ensure that the normalization is done by the correct number of total cells in the S1 cortex for each condition.
For figure 1D and 1E, the n=2 animals, and for control group no bar plotted. It should be increase the n number.
From the figure 1c example image, seems that it showed more c-fos expression than the control group.
For result part in figure 1g, the auhor mentioned that co-localization of c-Fos and calbindin significantly increased from 12.09 ± 1.2% to 38.7 ± 2.3% after burn injury (p = 0.021; Figure 1g), it is better to quatify the summary data here not only the example image.
In results part (section 2.2, at the end of the second paragraph, the "figure 1b" should be "figure 2b").
Inconsistent Abbreviations: The paper introduces several abbreviations (e.g., DEGs, GO, KEGG), it should be described in the first time, do not repeat them again。such as the burn injury, check this through the whole manuscript.
Author Response
Response to Reviewer #1:
Thank you very much for your thorough and constructive feedback. We appreciate your comments. In our revision, we have addressed all of your suggestions, concerns, and questions, and we are confident that, thanks to your valuable insights, we have significantly improved the manuscript.
We hope you will accept all of our responses, which you can read below.
Minor comments:
Comments 1: Figure 1d Ensure that the normalization is done by the correct number of total cells in the S1 cortex for each condition.
Response 1: For the normality test, the Shapiro-Wilk test yielded p-values above 0.76. This indicates that the data do not significantly deviate from a normal distribution, as the p-value is well above the typical significance threshold (i.e., 0.05). Thus, the assumption of normality, which is critical for a t-test, appears to be satisfied. However, considering that our sample size remained relatively small (with the number of slices ranging from 5 to 11), the t-test may still lack the statistical power to detect significant differences. Therefore, we chose a non-parametric alternative, the Mann-Whitney U test, which does not assume normality and is more robust for small sample sizes. We believe that these measures will enhance the reliability and validity of our data.
Comments 2: For figure 1D and 1E, the n=2 animals, and for control group no bar plotted. It should be increase the n number.
Response 2: We apologize for this shortcoming. Error bars were also included in the controls during the revision (see Figures 1d and e). Additionally, two more animals have been added to both experimental groups to increase the sample size as you recommended, allowing a re-evaluation of the percentage of cFos-immunoreactive (IR) nuclei and the percentage of co-localization between cFos and various interneuron subtypes.
To more accurately examine the lower limb region of the S1 cortex, we re-evaluated the older immunostained images alongside the newly stained ones and modified the evaluation criteria as follows: with the help of Imaris, we performed 3D cropping on the images, adjusting them to exact dimensions of 800 μm in length, 500 μm in width, and 35.5 μm in depth. This adjustment led to a reduction in the region of interest (ROI), specifically corresponding to the lower limb area of the S1 cortex under evaluation.
As a result, the number of immunoreactive spots and the colocalization values have been adjusted compared to those reported in the previous version of the manuscript. Please view pages 4-5 of the main text in the revised manuscript.
Additionally, the c-Fos fluorescence intensity values were not assessed, which is why this diagram from Figure 1 was excluded from the latest version of the manuscript and replaced with another diagram (see Figure 1e) that illustrates the percentage of colocalization between cFos and various interneuron subtypes. The figure legend for Figure 1 has been updated accordingly. Please take a look at it!
Comments 3: From the figure 1c example image, seems that it showed more c-fos expression than the control group.
Response 3: We apologize for any confusion regarding your question. The panels in Figure 1c have been updated with new confocal images, as we repeated the entire experiment, adding two additional mice per group. If your concern persists, could you please rephrase your question for further clarification?
Comments 4: For result part in figure 1g, the auhor mentioned that co-localization of c-Fos and calbindin significantly increased from 12.09 ± 1.2% to 38.7 ± 2.3% after burn injury (p = 0.021; Figure 1g), it is better to quatify the summary data here not only the example image.
Response 4: Agreed. The modifications have been made as recommended. With the addition of two more animals in each experimental group to increase the sample size, we re-evaluated the proportions of cFos-immunoreactive (IR) nuclei and the extent of co-localization between cFos and various interneuron subtypes. A new diagram has been included as Figure 1e. Please refer to Response 2 for more details.
Comments 5: In results part (section 2.2, at the end of the second paragraph, the "figure 1b" should be "figure 2b").
Response 5: Thank you for pointing this out. It has been modified accordingly.
Comments 6: Inconsistent Abbreviations: The paper introduces several abbreviations (e.g., DEGs, GO, KEGG), it should be described in the first time, do not repeat them again。such as the burn injury, check this through the whole manuscript.
Response 6: Thank you for bringing this to my attention. The entire manuscript has been reviewed and adjusted accordingly.
Reviewer 2 Report
Comments and Suggestions for Authors
Dear authors
The current manuscript investigated transcriptomic alterations in the S1 cortex of mice subjected to burn injury or formalin treatment, utilizing RNA sequencing (RNA-seq) one hour after injury. RNA-seq identified the downregulation of 82.4% of DEGs in burn injury and 32.4% in inflammatory pain with some overlapped DEGs. The authors suggested a significant difference in the cortical processing of pain with different origins. They concluded that transcriptomic remodeling in the S1 cortex is dependent on the sensory modality and the retrograde endocannabinoid network is activated during the acute pain response following burn injury. This point is very interesting for research however, I have some comments
- Abstract lacks subdivision so the methods and results are unclear or deficient
- Complete name should be mentioned before abbrev. for the 1st time.
- The number of the studied genes is unclear.
- The methods lack sample size, study design, and cited references in some parts
- The results lack the p-value in comparisons
- Discussion is very long
- Conclusion is not specific
- Other comments in the attached manuscript
Best Regards

Author Response
Response to Reviewer #2:
Thank you very much for your thorough and constructive feedback. We appreciate your insights. During the revision, we addressed all your comments, concerns, and questions, and we are confident that, with your helpful feedback, we have significantly improved the manuscript. Therefore, we hope you will find the revised version of the manuscript suitable for publication.
Here, you can find our responses to all your questions, one at a time:
Comments 1: Abstract lacks subdivision so the methods and results are unclear or deficient
Response 1: We appreciate your feedback and have revised the abstract as you requested suggested. However, the 200-word limit made it challenging to provide a more comprehensive explanation in the abstract.
Comments 2: Complete name should be mentioned before abbrev. for the 1st time.
Response 2: Thank you for pointing this out. The entire manuscript has been reviewed and modified accordingly.
Comments 3: The number of the studied genes is unclear.
Response 3: Agreed. As a result, an extra sentence has been added to the second paragraph of section 2.2 on page 5 as follows: “Using next-generation RNA-seq, a total of 13339 counts or sequences were identified in each experimental group. Based on the criteria (-1 < logFC < 1), we identified a total of 1116 differentially expressed genes (DEGs) in the S1 cortex of BI mice, while 136 DEGs were detected in FA mice compared to the untreated control sample.”
Additionally, another sentence has been added at the end of the first paragraph in section 2.3 on page 8 as follows.“Using the criteria of -0.5 logFC to 0.5, 4,619 DEGs were filtered in BI, while 988 DEGs were filtered in FA for GO analysis. ”
Furthermore, we have included additional tables to address this shortcoming (see Supplementary Table S4-S5).
Comments 4: The methods lack sample size, study design, and cited references in some parts
Response 4: We revised the main text regarding sample size and study design to address your valid concerns. The cited references you mentioned were removed while shortening the discussion. Please see our response 8 for further details.
Comments 5: The results lack the p-value in comparisons
Response 5: Agreed. Therefore, we have included extra supplementary tables that address this shortcoming (page 23). Supplementary Table S4: A list of BI-associated Gene Ontology (GO) terms and KEGG pathways, along with comprehensive bioinformatics data, including false discovery rate (FDR), p-values, and the number of involved genes, as well as other relevant details. Supplementary Table S5: A list of FA-associated Gene Ontology (GO) terms, along with comprehensive bioinformatics data, including the false discovery rate (FDR), p-values, and the number of involved genes, among other details. Please refer to the provided Supplementary Tables S4-S5 for more details.
Comments 6: Discussion is very long
Response 6: We agree and sincerely apologize for this. Consequently, we have shortened the discussion of the revised manuscript as much as possible during the revision process.
Comments 7: Conclusion is not specific
Response 7: It has been modified during the revision (on page 23). I hope the rewritten version will better meet the requirements of the conclusion.
Comments 8: Other comments in the attached manuscript
Response 8:
We have updated the abstract as requested.
The last paragraph of the introduction has been relocated to the end of the discussion.
References are cited on page 22 as recommended.
The sample size and study design were included (see the second paragraph of Section 4.1 on page 20), along with an additional sentence about the controls.
Section 4.2 has been revised for improved clarity.
The company and batch number for the Release have been added.
The references for immunofluorescent staining have been cited in Section 4.4 on page 21 (see line 8 of that section).

Reviewer 3 Report
Comments and Suggestions for Authors
This study investigated the transcriptomic alterations in the primary somatosensory cortex (S1) of mice after burn injury or formalin treatment, which is of significance for understanding the molecular mechanisms of pain processing. However, there are still some aspects that could be further improved.
Comments on the Quality of English LanguageThis study investigated the transcriptomic alterations in the primary somatosensory cortex (S1) of mice after burn injury or formalin treatment, which is of significance for understanding the molecular mechanisms of pain processing. However, there are still some aspects that could be further improved.
Author Response
Response to Reviewer #3:
Thank you very much for your thorough and constructive feedback. We appreciate your comments. In our revisions, we have addressed all of your comments, concerns, and questions. We believe that thanks to your valuable feedback, we have made significant improvements to the manuscript. Therefore, we hope you find the revised version of the manuscript acceptable for publication. Below, you can read our responses to all your questions:
Comment 1: In Figure 2: The Venn diagram and heatmaps are informative, but it would be better to provide more detailed information about the functional annotation of the overlapping and unique differentially expressed genes. For example, what are the key biological functions or signaling pathways that are specifically associated with the genes unique to each pain model?
Response 1: Figure 2 provides an overview of the RNA sequencing results. The inclusion of both the Venn diagram and heatmaps is essential for representing RNA sequencing data, even without indicating their biological function. This approach is standard for presenting data that has not yet undergone GO Enrichment Analysis. Supplementary Tables S1-S3 provide details about the molecular functions and biological processes related to the top 15 up- and downregulated DEGs. Additionally, several genes with the most significant upregulated or downregulated fold change values were discussed and contextualized within the current pain-related literature in the Discussion section.
Comment 2: In the section on Retrograde Endocannabinoid Signaling: The downregulation of some key genes in the retrograde endocannabinoid signaling pathway, despite the overall upregulation of the pathway, is an interesting finding. However, the authors only briefly mention potential reasons. More in - depth discussion on the possible regulatory mechanisms, such as post - transcriptional or post - translational regulation, would be valuable.
Response 2: We agree that understanding the potential regulatory mechanisms would be valuable. However, addressing this question is beyond the scope of the study without specific and direct experimental evidence. Additionally, one of the reviewers noted that the discussion is too lengthy and recommended shortening it. We must admit that it can be challenging to meet everyone's expectations. Balancing different perspectives and feedback is always a delicate task.
Comment 3: Regarding the Immunofluorescent Staining Results: The authors should discuss more comprehensively the implications of the changes in CB1 receptor and Ggamma14 protein levels. For example, how do these changes in protein levels relate to the functional changes in the S1 cortex, especially in the context of pain processing? And what are the potential interactions between CB1 receptor and Ggamma14 in the signaling pathways?
Response 3: We agree with the reviewer that this topic deserves its own paragraphs within the Discussion chapter. Please refer to it on pages 19-20.

Round 2
Reviewer 1 Report
Comments and Suggestions for Authors
The author has answered all of my questions, and no further revisions are needed.